# Reinforcement Learning with Simple Sequence Priors

**Tankred Saanum**[1†]     **Noémi Éltető**[1]     **Peter Dayan**[1,2]     **Marcel Binz**[1]     **Eric Schulz**[1]

[1]Max Planck Institute for Biological Cybernetics,   [2]University of Tübingen
[†]`tankred.saanum@tuebingen.mpg.de`

## Abstract

In reinforcement learning (RL), simplicity is typically quantified on an action-by-action basis – but this timescale ignores temporal regularities, like repetitions, often present in sequential strategies. We therefore propose an RL algorithm that learns to solve tasks with sequences of actions that are compressible. We explore two possible sources of simple action sequences: Sequences that can be learned by autoregressive models, and sequences that are compressible with off-the-shelf data compression algorithms. Distilling these preferences into sequence priors, we derive a novel information-theoretic objective that incentivizes agents to learn policies that maximize rewards while conforming to these priors. We show that the resulting RL algorithm leads to faster learning, and attains higher returns than state-of-the-art model-free approaches in a series of continuous control tasks from the DeepMind Control Suite. These priors also produce a powerful information-regularized agent that is robust to noisy observations and can perform open-loop control.

## 1   Introduction

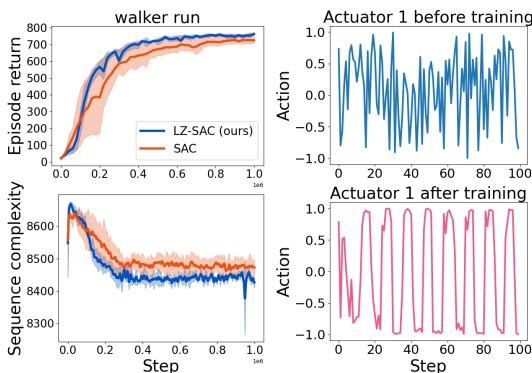

Figure 1: Action sequences produced by a bipedal walker become more compressible with learning. Our algorithm learns policies that solve tasks with simple action sequences, leading to decreased complexity and higher returns.

Simplicity is a powerful inductive bias [1–3]. In science, we strive to build parsimonious theories and algorithms that involve repetitions of the same basic steps. Simplicity is also important in the context of reinforcement learning (RL). Policies that are simple are often easier to execute, and practical to implement even with limited computational resources [4, 5]. Many control problems have solutions that are compressible: Motor behaviors like running and walking involve moving our legs in a periodic, alternating fashion (Fig. 1). Here it is the sequence of actions selected that is compressible. Sequences with repetitive, periodic elements are easier to predict and can be compressed more than sequences that lack such structure. In the current work, we augment RL agents with a prior that their action sequences should be simple: If solutions to control problems are generally compressible, one should consider only the set of *simple* solutions to a problem rather than the set

of *all* solutions. In a series of experiments, we show that RL with simple sequence priors produces policies that perform better and more robustly than state-of-the-art approaches without such priors.

37th Conference on Neural Information Processing Systems (NeurIPS 2023).

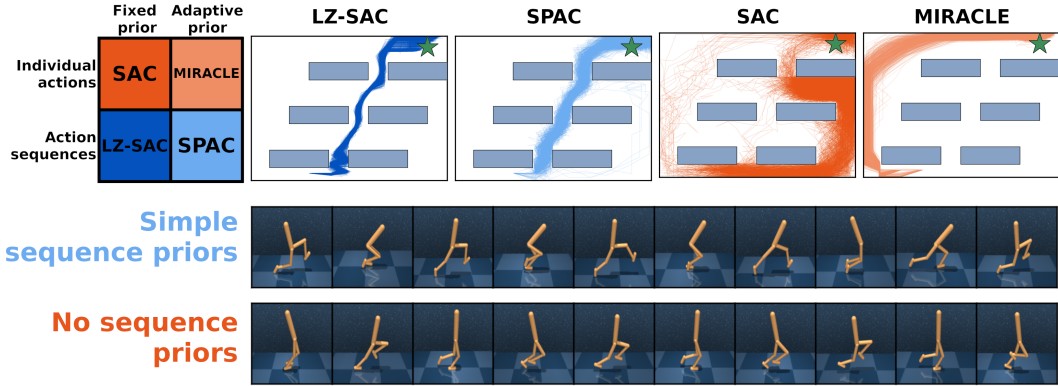

Figure 2: **Top left:** Policy regularization either incentivizes sequences or individual actions to be close to the prior. Priors may be distinct in that they stay fixed over training or change from episode to episode with learning. **Top right:** Agents need to navigate to a goal location, where the shortest path requires fine control, following a repeating pattern. After learning, SAC randomly diffuses among multiple paths. MIRACLE prefers a simple path that only goes up and then to the right. Since the optimal path is compressible (repeating UP and RIGHT in a periodic fashion), the agents with the simple sequence priors prefer this path. **Bottom:** An agent with simple sequence priors, in this case SPAC, learns simple strategies for walking, using mostly the left leg to push itself forward in a repetitive fashion.

Though there are methods for regularizing policies with respect to the *individual* actions they produce [6, 7], we present a method that explicitly regularizes the *sequences* of actions used to solve a task. Our regularization incentivizes the agent to use action sequences that can be compressed with a sequence prior. If an action sequence is likely under the prior, one needs fewer bits of information to represent it [8]. We explore two types of sequence priors: *i*) Priors in the form of an autoregressive sequence model [9, 10] that learns to predict future actions based on actions that were performed in the past and *ii*) priors distilled from a pre-programmed, lossless compression algorithm. Building on the Soft Actor-Critic algorithm (SAC) [7], we introduce Lempel-Ziv Soft Actor-Critic (LZ-SAC), using an off-the-shelf compression algorithm as its prior, and Soft Predictable Actor-Critic (SPAC), using a learned sequence prior (Fig. 2).

The contributions of this paper are the following: We introduce a model-free RL algorithm for maximizing rewards with simple action sequences. In a series of continuous control tasks, we evaluate the utility of such simple sequence priors. First, we investigate whether simple sequence priors speed up policy search: In our experiments, agents with simple sequence priors consistently outperform state-of-the-art model-free RL algorithms in terms of reward maximization. This holds both in terms of learning speed and often in the final performance. Our second result is that our regularization produces an information-efficient RL agent, using fewer bits of information to solve control problems. Information-regularized models are more robust and better at generalizing [11, 4, 12]. Lastly, we demonstrate the agents' advantages in environments with noisy and missing observations.[1]

## 2 Related work

The idea of simplicity has received significant attention in previous work. Maximum entropy RL, for instance, augments the reward function with an entropy maximization term, effectively encouraging the agent to stay close to a simple uniform prior policy over actions [13, 14]. Though uniform priors can lead to discontinuous and unpredictable behaviors, maximum entropy methods are considered simple in that they try to minimize the use of information about the state to select actions [4, 15, 16]. Many current approaches to deep RL – such as SAC [7] – rely on this principle. This concept has been further extended by models like Mutual Information Regularized Actor-Critic Learning (MIRACLE)

---

[1]For videos showing behaviors learned with our algorithm, see our project website:
https://sequencepriors.github.io
Code: https://github.com/tankred-saanum/simple_priors

[6] and others [17, 18], which use a learnable state-independent prior policy instead of the uniform prior assumed by SAC. SAC and MIRACLE both induce simplicity at the level of individual actions. In contrast, our proposed approach works on the level of action sequences.

It is not only possible to encode preferences for simplicity at the action level. Instead, simplicity can also be imposed by encouraging the agent to maintain simple internal representations – the core idea behind the information bottleneck principle [19]. Deep RL agents that rely on this principle have many appealing properties, such as improved robustness to noise, better generalization, and more efficient exploration characteristics [20–22]. Related to compression is predictability: Berseth et al. [23] learn a density model over states, and then learn a policy that seeks out states that are predictable, leading to self sustaining behaviors in unstable environments. On the opposite end there are methods that seek out unpredictable states [24, 25], or states that the agent *cannot* compress, to improve exploration. Recently, [4] demonstrated how to construct RL agents that learn policies that use few bits of information by not only compressing individual observations but entire sequences of observations. In some sense, our approach can be seen as a variant of the algorithm from [4]. However, we compress sequences of *actions*, rather than sequences of observations. Thus, our regularization does not target the complexity of the sequence of internal representations, but instead the complexity of the agent's behavior, manifested in the sequence of actions selected to solve a task.

Finally, simplicity is also an important feature of natural intelligence, where it has been repeatedly argued that simplicity is a unifying principle of human cognition [2]. For instance, [26] showed that people rely on compressed policies, ultimately leading to behavioral effects such as preservation or chunking [27, 28]. Likewise, [29] demonstrated that human exploration behavior can be described by RL algorithms with limited description length, while [30] showed that compression captures human behavior in a visual search task.

## 3 Control with simple sequences

In this section, we demonstrate how to construct RL agents that solve tasks using simple action sequences. We start by outlining the general problem formulation. We assume that the task can be posed as a Markov Decision Process (MDP). The MDP consists of a state space $\mathbf{s} \in \mathcal{S}$, an action space $\mathbf{a} \in \mathcal{A}$, and environment dynamics $p(\mathbf{s}_0)$ and $p(\mathbf{s}_{t+1} \mid \mathbf{s}_t, \mathbf{a}_t)$. The dynamics determine the probability of an episode starting in a particular state and the probability of the next state given the previous state and action, respectively. Lastly, there is the discount factor $\gamma$ and a reward function $r(\mathbf{s}_t, \mathbf{a}_t)$ that maps state-action pairs to a scalar reward term. The agent learns a policy $\pi_\theta(\mathbf{a}_t|\mathbf{s}_t)$ parameterized by $\theta$ that maps states to actions in a way that maximizes the sum of discounted rewards $\mathbb{E}_{\pi_\theta} \left[ \sum_{t=1}^{T} \gamma^t r(\mathbf{s}_t, \mathbf{a}_t) \right]$.

Though we want our RL agent to maximize rewards, we encourage it to do so with policies that produce simple action sequences. Inspired by previous approaches, we achieve this by augmenting the agent's objective [7, 4, 14], and search for a set of policy parameters $\theta$ that maximize reward while minimizing the *complexity* of the policy $C(\mathbf{a}_{t-\tau:t}, \mathbf{s}_t, \theta)$:

$$\max_\theta \mathbb{E}_{\pi_\theta} \left[ \sum_{t=1}^{T} \gamma^t (r(\mathbf{s}_t, \mathbf{a}_t) - \alpha \underbrace{C(\mathbf{a}_{t-\tau:t}, \mathbf{s}_t, \theta))}_{\text{Complexity cost}} \right] \tag{1}$$

where the hyper-parameter $\alpha$ controls the trade-off between complexity and discounted rewards.

We can recover various previous approaches using this formulation. If we, for instance, set $C(\mathbf{a}_{t-\tau:t}, \mathbf{s}_t, \theta) = \log \pi_\theta(\mathbf{a}_t \mid \mathbf{s}_t)$, we obtain maximum entropy RL algorithms such as SAC. SAC implicitly assumes a uniform prior over individual actions. An alternative to using the uniform prior in maximum entropy RL is to learn a parameterized prior over actions $p_\theta(\mathbf{a})$ based on the empirical distribution of actions the agent selects when solving the task [18, 6]. Setting $C(\mathbf{a}_{t-\tau:t}, \mathbf{s}_t, \theta) = \log \pi_\theta(\mathbf{a}_t \mid \mathbf{s}_t) - \log p_\theta(\mathbf{a}_t)$, we obtain MIRACLE.

## 3.1 Simplicity with learned priors

While both SAC and MIRACLE compress sums of *individual* actions, they do not account for the structure that is present in whole action sequences. To close this gap, we present two methods for regularizing policies on the level of action sequences. For the first, we train a prior distribution $\phi_\theta(\mathbf{a}_t \mid \mathbf{a}_{t-\tau:t-1})$ to predict the agent's future actions from actions it performed in the past. We parameterize the prior as a neural sequence model. We use a causal transformer model [9, 31] to parameterize $\phi_\theta$. Though any type of sequence model could be used in principle, Transformers are arguably better suited for learning complex sequence data with long-range dependencies. We can augment the reward function to incorporate the preference for predictable action sequences as follows:

$$\tilde{r}(\mathbf{s}_t, \mathbf{a}_{t-\tau:t}) = r(\mathbf{s}_t, \mathbf{a}_t) - \alpha(\log \pi_\theta(\mathbf{a}_t \mid \mathbf{s}_t) - \log \phi_\theta(\mathbf{a}_t \mid \mathbf{a}_{t-\tau:t-1})) \tag{2}$$

where $\mathbf{a}_{t-\tau:t-1}$ is a sequence of the last $\tau$ actions. Optimizing this objective, the agent will get rewarded for performing behaviors that the sequence model can predict better. The sequence model can learn to predict action sequences more easily if they contain structure and regularity. This has two interesting implications. *i*) The agent is incentivized to visit states where its actions will be predictable, for instance by oscillating between states in a periodic manner. *ii*) To perform actions that make it easier for the sequence model to predict future actions, for instance by performing behaviors that signal to the sequence model how it will behave in the future. We refer to this agent as the Soft Predictable Actor-Critic agent, or SPAC.

## 3.2 Simplicity with compression algorithms

Since the sequence model and the policy are adapting their behavior and prior towards each other, the augmented reward function will change throughout training. This plasticity can make it challenging to search for viable policies. Moreover, training a sequence model on top of the RL agent creates additional computational overhead. We, therefore, explore the possibility of instilling a simplicity preference without the use of a sequence prior that necessarily adapts over episodes.

This second method for distilling simple sequence priors relies on off-the-shelf data compression algorithms [32]. Lossless data compression algorithms like `LZ4`, `bzip2` and `zlib` encode data into sequences of symbols from which the original data can be reconstructed or decompressed exactly. If there are repetitions, regularities, or periodicity in the data, the length of the encoded sequence can be significantly shorter than the original size of the data (Fig. 3). Relying on pre-programmed rules for data compression, this simplicity prior will not change over the course of training. Since compression

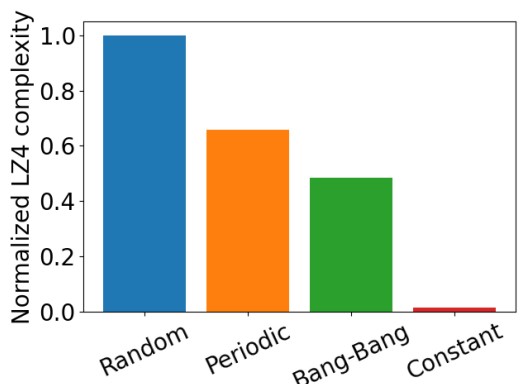

Figure 3: Some sequences are more compressible than others. A sequence of randomly generated numbers is less compressible than sequences with periodicity, sequences that only contain two types of values (also known as Bang-Bang control), or constant sequences that only contain a single number.

algorithms like `LZ4` are fast, the sequence prior can be implemented with little computational overhead.

In this setting, we compute $C$ using the extra number of bits needed to encode $\mathbf{a}_t$ given that we have already encoded $\mathbf{a}_{t-\tau:t-1}$:

$$\delta_t = \text{len}(g(\mathbf{a}_{t-\tau:t-1})) - \text{len}(g(\mathbf{a}_{t-\tau:t})) \tag{3}$$

$$\tilde{r}(\mathbf{s}_t, \mathbf{a}_{t-\tau:t}) = r(\mathbf{s}_t, \mathbf{a}_t) - \alpha(\log \pi_\theta(\mathbf{a}_t \mid \mathbf{s}_t) - \delta_t) \tag{4}$$

where $g(\cdot)$ is our compression function and len$(\cdot)$ returns the length of a sequence. We use the `LZ4` compression algorithm to compute the augmented rewards and refer to this agent as the LZ-SAC agent.

### 3.3 Implementational details

We implement all agents as extensions of the SAC algorithm. SAC is an off-policy actor-critic algorithm that performs maximum entropy RL. We train critics to learn the augmented $Q$-value function $\tilde{Q}(\mathbf{s}_t, \mathbf{a}_{t-\tau:t}) = \mathbb{E}[\sum_{t=1}^{N} \gamma^t \tilde{r}(\mathbf{s}_t, \mathbf{a}_{t-\tau:t})]$ with temporal-difference learning [33]. The actors and sequence models are trained to minimize the same loss:

$$\mathcal{L} = \mathbb{E}_{\mathbf{s}_t, \mathbf{a}_{t-\tau:t} \sim \mathcal{D}}[\alpha(\log \pi_\theta(\mathbf{a}_t \mid \mathbf{s}_t) - \log \phi_\theta(\mathbf{a}_t \mid \mathbf{a}_{t-\tau:t-1})) - \tilde{Q}(\mathbf{s}_t, \mathbf{a}_{t-\tau:t})] \tag{5}$$

where $\mathcal{D}$ is a replay buffer and $\mathbf{a}_t \sim \pi_\theta(\cdot \mid \mathbf{s}_t)$. The LZ-SAC actor minimizes the same loss except that $-\log \phi_\theta(\mathbf{a}_t \mid \mathbf{a}_{t-\tau:t-1})$ is replaced with the term in Eq. 3. In practice, we take the minimum of two target $Q$-networks to train the actor and critic. Learning is achieved by sampling experiences from a replay buffer. To calculate the augmented rewards, we simply sample action sequences $\mathbf{a}_{t-\tau:t-1}$ that led to the $(\mathbf{s}_t, \mathbf{a}_t, \mathbf{s}_{t+1}, r_t)$ tuple used for training (see Appendix A for full implementational details).

## 4  Simple sequence priors guide policy search

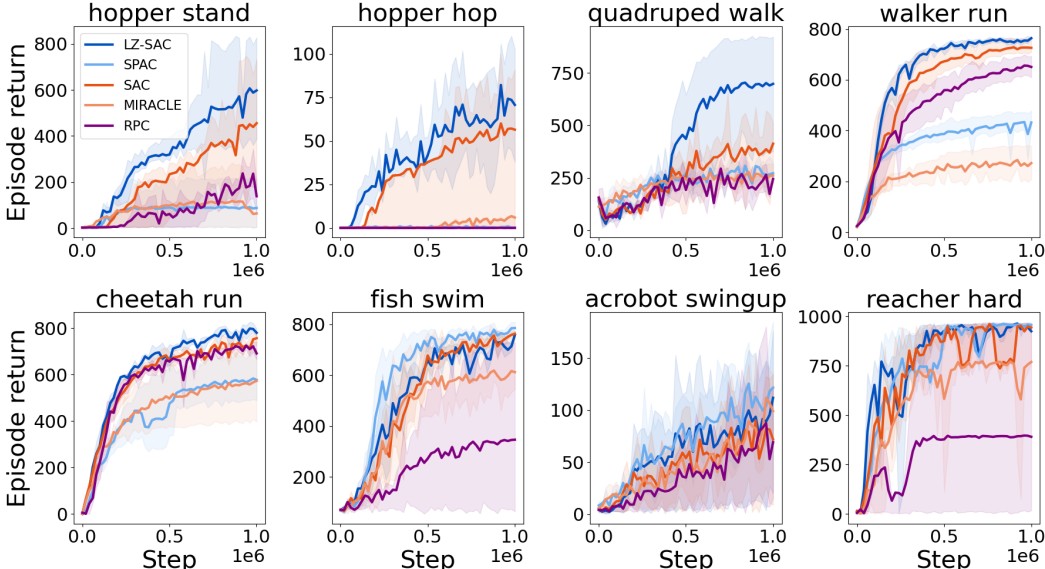

Figure 4: Learning curves of agents in the DeepMind Control suite. Overall, LZ-SAC shows the best learning speed and final performance. Lines are the average episodic returns collected in 20 test episodes with a deterministic policy, averaged over five agents trained with different seeds. Shaded regions represent 20-80 performance percentiles.

We evaluated the agents described in Section 3 on eight continuous control tasks from the DeepMind Control Suite [34]. As an additional baseline we included Robust Predictable Control (RPC) from [4], which compresses sequences of states rather than actions. Many of the tasks in the DeepMind Control Suite promote behaviors with periodic action sequences, such as running and walking. While specialized architectures exist for such tasks [35], we expect compressibility to be a useful inductive bias for learning these behaviors. We trained agents for 1 million environment steps across five seeds and evaluated their abilities at regular intervals with a deterministic policy, as in [7, 36]. We tuned $\alpha$ for each agent and found an $\alpha = 0.1$ to give the best performance in almost all tasks. We found lower information costs $\alpha$ worked better for RPC. Tasks and hyperparameter fitting is described in Appendix B.

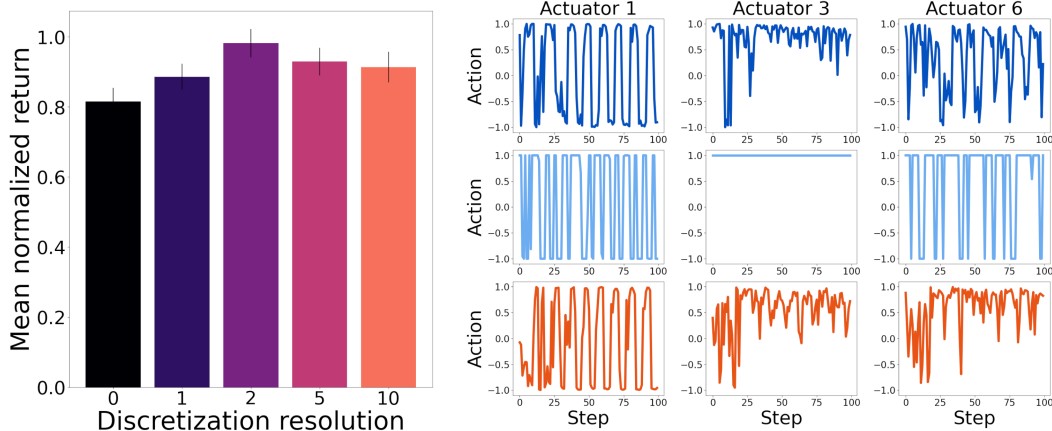

Figure 5: **Left**: LZ-SAC works best when the discretization resolution allows the agent to distinguish between compressible and incompressible sequences. Bars show mean return and SEM from 20 evaluation episodes over three seeds and three environments after 300k steps. **Right**: Time series of actions produced by the LZ-SAC, SPAC and SAC agent in the `walker run` task. Action time series of the SPAC agent exhibits a simpler periodic pattern, even outputting a constant value for its third actuator. Actuators were chosen to show qualitatively different behaviors.

In a majority of the tasks, the LZ-SAC agent outperformed the SAC, RPC and MIRACLE agents in learning speed and often final performance (Fig. 4). At worst, the LZ-SAC agent matched the learning curves of SAC. This suggests that learning policies with simple sequence priors is indeed fruitful for policy search. We investigated whether this performance difference could simply be attributed to LZ-SAC acting more deterministically than SAC. Lowering the incentive of acting randomly for SAC did not close the performance gap, and often led to worse returns (see Appendix B.1).

We conducted an additional ablation experiment to make sure the performance gain could be attributed to the compressibility incentive: We varied the resolution with which we discretized the action sequences used as input to the compression algorithm when LZ-SAC was trained. Across three DeepMind Control tasks (`cheetah run`, `acrobot swingup` and `walker walk`), we observe that both too low and too high resolutions remove the performance gain of LZ–SAC: When the resolution is 0, the compression bonus no longer conveys a signal about the simplicity of the policy. Conversely, if the resolution is too high, every action sequence is equally incompressible due to the continuous nature of the action space. We find that rounding to two decimal places gave the best performance on average across the tasks (see Fig 5, left). Though we do not expect LZ-SAC to always outperform SAC in more generic control settings, we see an improvement in many tasks with periodic elements, like walking and running. We evaluate LZ-SAC in non-periodic tasks in Section 8.

In two tasks, `acrobot swingup` and `fish swim`, the SPAC agent showed a competitive advantage over the other models. However, the SPAC agent lagged behind both the LZ-SAC agent and SAC agent in tasks from the `hopper`, `cheetah`, and `walker` domains. Here the policy that the SPAC agent learned achieves roughly 75% of the return of the LZ-SAC agent.

The policies learned by the SPAC agent shine in a different setting: The agent has discovered solutions to these tasks that essentially use fewer action dimensions than the competitors (Fig. 5, right): For certain actuators $a_i$, the agent outputs a constant value throughout the episodes. For other actuators, the agent alternates between two extreme values, like a soft bang-bang controller [37, 38]. Essentially, the SPAC agent figures out which degrees of freedom it can eliminate without jettisoning rewards. Having fewer degrees of freedom makes it easier to predict the action sequences produced by the policy. This suggests that policy compression using adaptive sequence priors is better suited in tasks with low-dimensional action spaces. Lastly, the difficulties of learning a policy and a sequence prior jointly can be mitigated by using a pre-trained sequence model as a prior. We pre-trained Transformer models to predict action sequences produced by the converged LZ-SAC agents in all eight control tasks. Using the pre-trained Transformers with frozen weights sped up learning significantly and allowed the SPAC agent to learn more rewarding behaviors (see Appendix G).

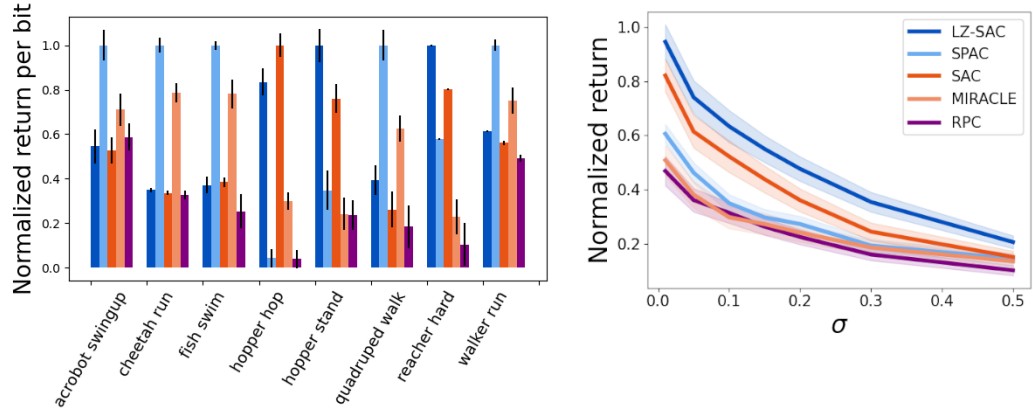

Figure 6: **Left**: Normalized return per bit attained by the agents in the eight tasks. Agents with simple sequence priors achieve better return per bit ratios. Error bars represent the standard error of the mean (SEM). **Right**: Normalized return averaged over all tasks as a function of noise scale. Error bands represent the SEM.

## 5 Simple sequence priors for information-regularized RL

The expected difference in log-likelihood of the agents' actions under the policy versus the prior is an upper bound on the mutual information between states and actions [4, 11, 19]. Encouraging this difference to be low acts as an information-regularizer, the prior $p(\mathbf{a}_t|\mathbf{a}_{t-\tau:t-1})$ being the information bottleneck. We tested the information-efficiency of learned policies; that is, how much reward the agents could collect relative to the information they used to make decisions. For the experiments, we again tested the deterministic versions of the agents. Simulating 25 episodes, we computed how much reward the agents were able to collect divided by the entropy of the distribution of actions used to solve the task $\mathbb{E}\left[\frac{\sum_{t=1}^{T} r_t}{\mathcal{H}[\mathbf{a}]}\right]$ (see Appendix D.1 for details and experiments with stochastic policies). Since the policies were deterministic, this entropy term approximated the mutual information between states and actions $I(\mathbf{s};\mathbf{a})$ (see Appendix D). In the left panel of Fig. 6, we show the normalized episodic return per bit. This quantity represents how much reward the agent attains per bit of information it uses on average to make a decision over the course of the episode.

The SPAC agent attained a superior return per bit ratio in five out of eight tasks. LZ-SAC attained the highest return per bit ratio in two tasks, and SAC in one. This indicates that action sequence compression is a powerful information-regularizer, allowing agents to find policies that use significantly fewer bits of information to collect reward than both policy compression models (SAC and MIRACLE), and state sequence compression models (RPC).

## 6 Robustness to noise

Information-regularized policies tend to show stronger robustness to noisy observations [39, 4]: The less an agent's actions vary systematically with the state, the less will a perturbation to the agent's observation affect its actions. We assessed how observation noise affected the agents' ability to collect rewards. In the following experiments, we added Gaussian noise to the observations the policies were conditioned on, $\mathbf{s}_t \leftarrow \mathbf{s}_t + \epsilon_t$ where $\epsilon_t \sim \mathcal{N}(\mathbf{0}, diag(\boldsymbol{\sigma}))$. We tested the agents on a series of noise levels $\sigma_j \in [0.01, 0.05, 0.1, 0.15, 0.2, 0.3, 0.5]$. The effect of noise was probed in all tasks except the `hopper hop` task, since here only the LZ-SAC agent reliably learned a policy that was better than random. Each agent was evaluated using 50 episodes for each noise level.

We evaluated the agents based on how much reward they collected given various levels of observation noise. Averaged over all tasks, the LZ-SAC agent showed the best ability to collect rewards when observations were perturbed with Gaussian noise (see the right panel in Fig. 6). The agents that were better at maximizing rewards showed greater sensitivity to noise: compared to the noise-free setting, LZ-SAC and SAC dropped to 20% and 17% of their average performance, respectively. While the

LZ-SAC agent suffered greater percentage drops in return than the MIRACLE and SPAC agents, it still retained the highest performance for all noise levels. In the highest noise settings, SAC is comparable to the MIRACLE and SPAC agents, despite its generally stronger performance in the noise-free setting. This indicates that the LZ-SAC agent performed better in the noisy setting not only because the policy it learned was *generally* better at maximizing rewards, but also because of robustness properties afforded by the sequence prior.

# 7   Open-loop control

If simple action sequences are pervasive in policies learned with RL, these priors could provide a good starting point for policy search. To further test this claim, we evaluated how well tasks from the DeepMind Control Suite could be solved by autoregressively generated action sequences from the sequence priors themselves. We omitted RPC from this analysis since it has the same prior policy as SAC. In our experiments, all agents produced the first 15 actions of an episode in a closed-loop manner. We then conditioned the sequence priors with these first 15 actions and sampled actions autoregressively for the remainder of the episode. The priors of the SAC and MIRACLE agents have no autoregressive component, and generated action sequences in a memory-less manner. We approximated samples from the `LZ4` prior by discretizing the action space and sampling the next action proportionally to how low its encoding cost is, given the previous actions.

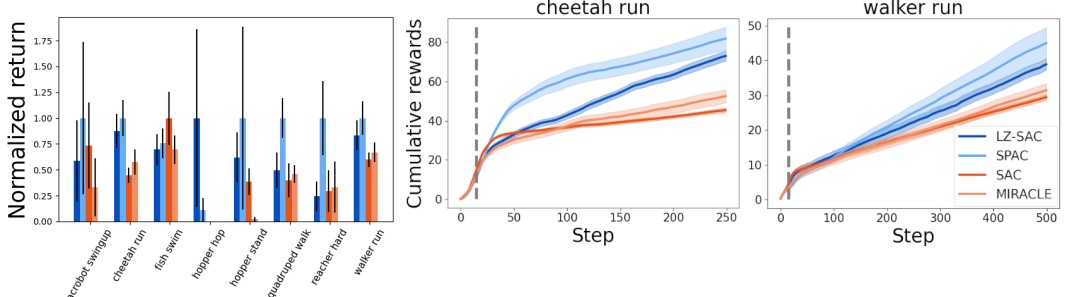

Figure 7: **Left**: Bars represent return attained in the open-loop phase exclusively. Error bars represent the SEM. The sequence prior learned by the transformer generally performs the best. Notably, the `LZ4` prior performs well in tasks solved with periodic action sequences, like `cheetah` and `walker`. **Right**: Average cumulative reward obtained by agents in the `cheetah` and `walker` tasks. Dashed lines indicate where the open-loop controls start.

The adaptive prior implemented as a transformer generally performs the best in the open-loop setting (Fig. 7, left). This is expected, as it was trained to predict behaviors that solve the tasks. In the `fish swim` task a uniform prior collects more rewards in the open-loop phase than the sequences generated by the transformer. However, increasing the number of closed-loop actions used to prompt the transformer to 25 made it surpass the performance of the uniform prior (Appendix E). This points to the importance of providing the sequence models with sufficient context to allow them to accurately predict behavior.

More interesting is the performance of the prior obtained from the `LZ4` algorithm. Not only does it perform better than chance, but even comes close to the performance of the learned sequence prior in tasks like `cheetah run` and `walker run`. By conditioning on only a few actions from the policy, autoregressively approximating samples from `LZ4`'s prior produced behaviors outperforming the non-sequential priors used by SAC and MIRACLE (Fig. 7, right). This vindicates the compressibility prior as a starting point for policy search.

# 8   Non-periodic and high-dimensional environments

Many DeepMind Control Suite tasks have solutions that are composed of repeating sub-sequences. Can simple sequence priors be beneficial in tasks without prevalent periodic aspects, or in more complex tasks with high-dimensional state spaces? We first evaluated LZ-SAC against SAC on

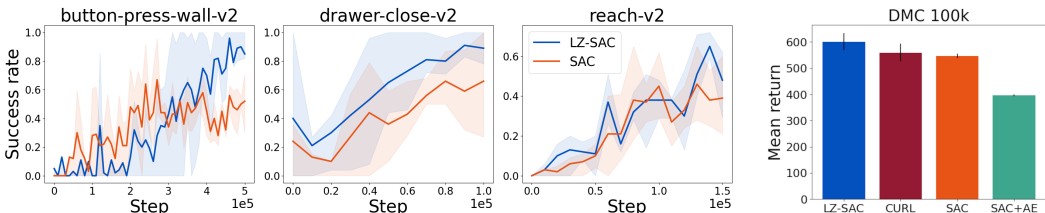

Figure 8: **Left**: Learning curves for three robotic manipulation tasks in the Metaworld benchmark. Lines represent the success rate across 20 test episodes with a deterministic policy, averaged over five agents trained with different seeds. **Right**: Average return attained in the pixel-based version of the Deepmind Control Suite tasks across 20 test episodes after 100k environment steps, averaged over six tasks and five seeds. Error bars reflect standard deviation. LZ-SAC with image augmentation outperforms the SAC baseline and two state-of-the-art off-policy methods that combine SAC with representation learning: one using contrastive learning (CURL) and one using an autoencoder (SAC+AE).

three tasks from the Metaworld benchmark [40]. The Metaworld benchmark consists of robotic manipulation tasks where periodic action sequences are less prevalent. Despite this, we found that the simple sequence priors allowed agents to learn policies with higher success rates faster (see Fig. 8, left). In fact, the solutions LZ-SAC developed for the robotics tasks often consisted of single, smooth movements with the Sawyer arm. SAC on the other hand relied on more convoluted movements to manipulate the environment, which were more prone to failure.

Next, we benchmarked LZ-SAC in pixel-based versions of six DeepMind Control Suite tasks. We trained LZ-SAC and a SAC baseline on 100k environment steps, and modified both algorithms with a convolutional neural network encoder and performed a random shift augmentation to images before training the actor and critic. With this simple modification and our compression bonus, LZ-SAC outperformed state-of-the-art off-policy algorithms like CURL [41] and SAC+AE [36] on average over the six tasks (see Fig. 8, right). Across five seeds, LZ-SAC outperformed the three baseline models on three out of the six tasks. See Appendix C.1 for the full scores and implementation details.

# 9   Discussion

We have argued that simplicity is a powerful principle to guide policy search in RL tasks. Because control problems are often solved with sequences of actions that contain repeating temporal patterns, we proposed to use simple sequence priors to create effective and robust RL agents. To provide agents with a notion of compressibility, we proposed two models: One where the strategy used for compression was fixed throughout training (LZ-SAC), and one where the strategy itself could change with experience (SPAC). While the LZ-SAC agents either outperformed or matched the performance of state-of-the-art methods like SAC, the SPAC agents learned more compressible strategies, attaining more rewards while using fewer bits of information to make a decision. Furthermore, agents trained with the LZ-SAC algorithm proved to be the most robust to observation noise. Lastly, both the trained transformer model and the prior distilled from the `LZ4` algorithm could autoregressively generate rewarding behaviors in continuous control tasks.

While SPAC showed a better ability to maximize rewards than MIRACLE, returns were lower than SAC and our alternative regularization technique. This is not unexpected. The transformer always required some amount of learning to be able to predict a particular action sequence. The `LZ4` algorithm, on the other hand, could immediately provide feedback about the compressibility of the agent's action sequences without any learning. For SPAC, having to learn a sequence prior induced a stronger bottleneck, resulting in more compressed policies. This is consistent with results reported by Eysenbach et al. [4], where a learned dynamics model was used to compress sequences of states: Here compression with a learned prior led to lower returns, but a higher return per bit rate. Our results suggest that sequence compression based on off-the-shelf compression algorithms like `LZ4` are better for policy search since there is no need for learning a sequence prior from scratch.

**Limitations:** Action sequence compression requires either an adaptive prior, a neural sequence model, or a pre-programmed compression algorithm. The particular algorithm used for compression adds computational overhead and determines the types of action sequences that will be favored by the agent [32]. Future work should address the ways in which different compression algorithms or sequence priors affect policy regularization. Furthermore, a sufficiently sophisticated sequence model could in principle learn to predict complex action sequences. A possible extension of our work could be to further penalize the description length of the weights of the sequence model, or the compression algorithm, itself [42]. Finally, while we evaluated our algorithm on a large and diverse set of control tasks within the DeepMind Control Suite and Metaworld, the utility of simple sequence priors could be tested on other benchmarks. In discrete action settings, Atari games [43] would be an appropriate benchmark.

**Future Directions:** A central feature of simple action sequences is that they are predictable. Being able to predict one's future behavior from past behavior could allow agents to simplify and compress their representations of the state of the world [11, 4]: If the point of observing the state is to determine what action to choose, one could discard information about the state of the world simply by considering the actions that were performed previously. This suggests that simple sequence priors could be beneficial for compressing policies and internal representations jointly.

Finally, humans show a preference for simplicity and compressibility in various domains [30, 2]: We not only produce art and music full of patterns and regularity [44], but also explore novel environments using compressible trajectories [45], and rely on simple rules to explain and generalize about complex stimuli relationships [46–48]. Recently, dopamine activity in the tail of the mouse striatum was argued to encode an action prediction error signal [49]. Such a signal also features in our augmented reward function to compress the policy. In the end, our algorithms could therefore serve as models of how biological agents learn compressible sequential strategies from reinforcement.

## Acknowledgements

We thank the four anonymous reviewers for the instructive and helpful feedback during the review process, as well as the members of the Computational Principles of Intelligence Lab for feedback provided throughout the project. We also thank Can Demircan for providing comments on an earlier draft. This work was supported by the Max Planck Society, the German Federal Ministry of Education and Research (BMBF): Tübingen AI Center, FKZ: 01IS18039A, and funded by the Deutsche Forschungsgemeinschaft (DFG, German Research Foundation) under Germany's Excellence Strategy–EXC2064/1–390727645.

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

# A  Implementation details

Our algorithm is an extension of the Soft Actor-Critic algorithm [7, 50], implemented in PyTorch. Like [36, 51], we initialize agents' replay buffer with a 1000 seed observations collected with a uniform random policy. We update the Q-network pairs to predict the augmented $Q$ value function at every interaction step, and the actor network to maximize the augmented $Q$ value function every second interaction step. The augmented $Q$ targets take the following form:

$$y = r(\mathbf{s}_t, \mathbf{a}_t) + \gamma\left[Q(\mathbf{s}_{t+1}, \mathbf{a}_{t+1}) - \alpha(\log \pi_\theta(\mathbf{a}_{t+1}|\mathbf{s}_{t+1}) - \log \phi_\theta(\mathbf{a}_{t+1}|\mathbf{a}_{t-\tau:t}))\right] \qquad (6)$$

where $\mathbf{a}_{t+1} \sim \pi_\theta(\cdot|\mathbf{s}_{t+1})$. To train the networks we sampled $B$ tuples of state, actions, next state, reward and terminal flags, as well as the $\tau$ actions that led to them. To allow the agent to train on observations early in episodes, we sampled $\tau$ from a uniform distribution of integers between 5 and $\tau_{\max}$ (see tables 1, 2) for every mini-batch sample used for training. We update the adaptive priors used in SPAC and MIRACLE together with the actor network. All actor and critic networks consisted of two hidden layers with 256 ReLU units [52] each. The action prior used in MIRACLE was implemented as a multivariate isotropic Gaussian with learnable mean and standard deviation.

Since actions are bounded between -1 and 1, we transform actions sampled from the policy using the $\tanh$ transform $\mathbf{a}_t = \tanh(\mathbf{u}_t)$, $\mathbf{u}_t \sim \pi_\theta$. We transformed the log-likelihood of an action under a Gaussian policy $\pi_\theta$ or action prior $\phi_\theta$ using the following formula [7, 50]:

$$\log \pi_\theta(\mathbf{a}_t|\mathbf{s}_t) = \log \mu(\mathbf{u}_t|\mathbf{s}_t) - \sum_{i=1}^{D} \log(1 - \tanh^2(u_i)) \qquad (7)$$

$$\log \phi_\theta(\mathbf{a}_t|\mathbf{a}_{t-\tau:t-1}) = \log \psi_\theta(\mathbf{u}_t|\mathbf{a}_{t-\tau:t-1}) - \sum_{i=1}^{D} \log(1 - \tanh^2(u_i)) \qquad (8)$$

where $\log \psi_\theta(\mathbf{u}_t|\mathbf{a}_{t-\tau:t-1})$ is the log likelihood of the untransformed action $\mathbf{u}_t$ under the untransformed sequence prior $\psi_\theta$.

## A.1  Quantifying compressibility

We used the `LZ4` algorithm to quantify the compressibility of action sequences. The following code snippet describes how we computed the sequence complexity term in Eq. 3

```
sequence_i = action_sequence.numpy().ravel()
# get length of compressed sequence at t
length_t1 = len(compression_algorithm.compress(sequence_i))
action_next = policy(state).numpy().ravel()   # get next on-policy action
sequence_j = np.concatenate((sequence_i, action_next), axis=0)
# get length of compressed sequence at t+1
length_t2 = len(compression_algorithm.compress(sequence_j))
delta = length_t - length_t2   # delta is the difference
```

Since the actions were continuous vectors, we quantized all action sequences with the following function:

```
def quantize(action_sequence, N=100):
    return (action_sequence*N).floor()
```

Here $N$ determines the granularity of the quantization, with lower $N$ producing more coarse-grained sequences. We set $N = 100$ for our experiments.

## A.2  Pseudo-code for `lz4` algorithm

The `lz4` algorithm compresses sequences by replacing repeating sub-sequences in the data with references to an earlier occurring copy of the sub-sequence. These copies are maintained in a sliding

window. Repeating sub-sequences are encoded as *length-distance* pairs $(l, d)$, specifying that a set of $l$ symbols have a match $d$ symbols back in the uncompressed sequence. The following pseudo-code sketches compression implemented by `LZ4` [32]:

---

**Algorithm 1** LZ4 pseudo-code

---

**Require:** Buffer size $b$, window size $w$, sequence $\mathbf{k}$
  $t = 0$
  window $\leftarrow \langle \, \rangle$
  **while** t $<$ len($\mathbf{k}$) **do**
     match $\leftarrow$ longest repeated occurrence in window found in $\mathbf{k}_{t:t+b}$
     **if** match exists **then**
        $d \leftarrow$ distance to start of match
        $l \leftarrow$ length of match
        $c \leftarrow$ symbol at $\mathbf{k}_{t+l}$
     **else**
        $d \leftarrow 0$
        $l \leftarrow 0$
        $c \leftarrow 0$
     **end if**
     output $(d, l, c)$
     start $\leftarrow \max(t - w + l, 0)$
     end $t \leftarrow t + l$
     window $\leftarrow \mathbf{k}_{\text{start:end}}$
     $t \leftarrow t + l + 1$
  **end while**

---

# B Hyperparameters

## B.1 Increasing reward scale

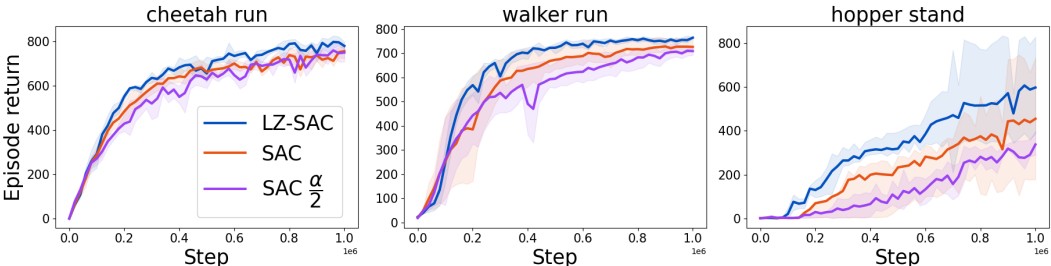

Figure 9: Halving the incentive of acting randomly does not close the performance gap between LZ-SAC and SAC.

We investigated whether LZ-SAC's performance improvement could simply be attributed to the incentive to act more deterministically. We tested whether we could attain the same level of performance with SAC just by lowering the incentive of acting randomly. Doubling the scale of extrinsic reward relative to the intrinsic reward of acting randomly did not close the gap between the algorithms (Fig. 9). Instead, we see a decline in performance when we set the incentive of randomness lower than $\alpha = 0.1$ (or $\alpha = 0.02$ in `walker run`). This indicates that there is value in having a preference for simplicity on the sequence level that goes beyond simply being predictable at the level of individual actions.

## B.2 Algorithm hyperparameters

We implement all algorithms using hyperparameters from [36], with slight deviations depending on the task. Since the computational overhead of using `lz4` as a compressor is small compared to the

transformer, we train the agents using larger action sequences. All networks were trained with the Adam optimizer [53]. A full list of hyperparameters is given below:

Table 1: Hyperparameters used for SAC, MIRACLE, LZ-SAC, and SPAC

| Hyperparameter | Value |
|---|---|
| Complexity cost $\alpha$ (`walker run`, `hopper hop`, `quadruped walk`) | 0.02 |
| Complexity cost (all other environments) $\alpha$ | 0.1 |
| Discount $\gamma$ | 0.99 |
| Critic update frequency | 1 |
| Actor update frequency | 2 |
| Action prior update frequency | 2 |
| Soft update $\rho$ | 0.01 |
| Batch size | 128 |
| Learning rate actor | $10^{-3}$ |
| Learning rate critic | $10^{-3}$ |
| Optimizer | Adam |
| Max context length LZ-SAC ($\tau_{\max}$) | Interaction steps in episode $\times$ 0.4 |
| Max context length LZ-SAC ($\tau_{\max}$; `walker run`) | Interaction steps in episode $\times$ 0.25 |

## B.3 Transformer

The SPAC agent uses a causal transformer [9] to learn a prior over action sequences. Our transformer was implemented with the following hyperparameters:

Table 2: Transformer hyperparameters.

| Hyperparameter | Value |
|---|---|
| Attention heads | 5 |
| Embedding dimensions | 30 |
| Learning rate decay | Linear |
| Warmup tokens | 10000 |
| Max context length ($\tau_{\max}$) | 20 |
| Number of layers | 2 |
| Learning rate | $3 \times 10^{-4}$ |
| Dropout | 0.1 |
| Optimizer | Adam |

## C   Task specification

We evaluated agents on tasks from the DeepMind Control Suite. Though dynamics are otherwise deterministic, the starting state of an episode is sampled from a distribution $p(\mathbf{s}_0)$. All episodes consist of 1000 environment steps. However, in practice the episode length is reduced to a number of *interaction steps*, that is smaller than 1000. This is due to an action repeat hyperparameter which determines how many times an action $\mathbf{a}_t$ is repeated after it is selected. An action repeat value of 4 thus reduces the number of time steps where the agent needs to act to 250 interaction steps. The action repeat hyperparameter makes it more practical to train agents in the DeepMind Control Suite [34]. We adopt conventional action repeat settings from the literature [36]. In the `walker` and `hopper` domains we fitted the action repeat value for all agents among $[2, 4, 8]$ and chose the value that produced the best performance. Table 3 shows the action repeat values used in our experiments:

### C.1   Pixel-based control

Our LZ-SAC and SAC implementations in the visual control domain differed little from the state-based implementations. We equipped the agents with the convolutional neural network architecture and image augmentation transformation from [54]. All MLPs had two hidden

Table 3: Action repeat values.

| Task | Action repeat |
|---|---|
| acrobot swingup | 8 |
| cheetah run | 4 |
| fish swim | 4 |
| hopper hop | 8 |
| hopper stand | 8 |
| quadruped walk | 4 |
| reacher hard | 4 |
| walker run | 2 |
| walker run (SPAC) | 4 |

Table 4: All final average model scores in the DeepMind Control 100k benchmark with pixel observations. Scores are averaged over 10 runs after 100k steps for five seeds. Scores of the other baselines are the ones reported in the respective papers [41] [36].

| DMC 100k | LZ-SAC | SAC | CURL | SAC+AE |
|---|---|---|---|---|
| Finger, Spin | **814** | 738 | 767 | 740 |
| Cartpole, Swingup | **683** | 609 | 582 | 311 |
| Walker, Walk | **635** | 609 | 403 | 394 |
| Ball In Cup, Catch | 653 | 499 | **769** | 391 |
| Cheetah, Run | 307 | **396** | 299 | 274 |
| Reacher, Easy | 513 | 427 | **538** | 274 |

layers and 512 ReLU units. We optimized the $\alpha$ hyperparameter both for the LZ-SAC and SAC agents for all tasks with a grid search. For tasks with higher dimensional action spaces a lower $\alpha$ of 0.01 worked best. In the end, the best performing $\alpha$ for both algorithms was the same across task. Since LZ4 encoding lengths are not necessarily on the same scale as the log likelihoods of the actions given the state in Equation 4, we experimented with scaling the LZ4 encoding cost by 0.5, which slightly improved performance. The full scores of the models are in Table 4.

Table 5: Action repeat values and complexity cost for both LZ-SAC and SAC agents in the pixel-based tasks.

| Task | Action repeat | $\alpha$ |
|---|---|---|
| finger spin | 2 | 0.01 |
| cartpole swingup | 8 | 0.1 |
| walker walk | 2 | 0.01 |
| ball in cup catch | 4 | 0.01 |
| cheetah run | 4 | 0.01 |
| reacher easy | 2 | 0.1 |

## D  Mutual information approximation

The mutual information $I(X; Y)$ between variables $X$ and $Y$ is a measure of how much they depend on each other. In our case we are interested in the mutual information between states and actions $I(\mathbf{s}; \mathbf{a})$. The mutual information here quantifies how many bits of information knowing the outcome of the random variable $\mathbf{s}_t$ provides about the other random variable $\mathbf{a}_t$, in other words, how much the state reveals about what action will be selected. The more an agent's actions vary as a function of the state, the more bits of information the state reveals about the action that the agent will select.

The mutual information is defined as the following quantity

$$I(\mathbf{a}; \mathbf{s}) = \mathcal{H}[\mathbf{a}] - \mathcal{H}[\mathbf{a}|\mathbf{s}] \tag{9}$$

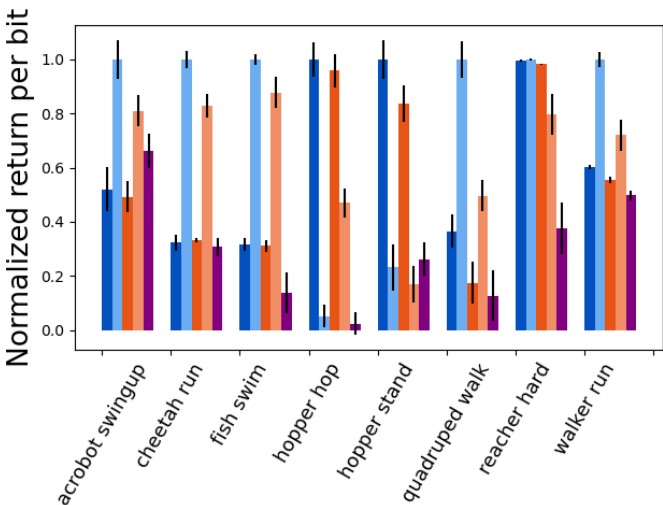

Figure 10: Return per bit for the stochastic policies. LZ-SAC attains the highest return per bit ratio in most tasks.

To compute the mutual information we need to know the entropy of the distribution of actions used to solve the task, and the conditional entropy of $\mathbf{a}|\mathbf{s}$. Since we make the policies deterministic, we know that $\mathcal{H}[\mathbf{a}|\mathbf{s}] = 0$. This way the mutual information reduces to the entropy over actions $\mathcal{H}[\mathbf{a}]$. Given a sample of actions produced by the agent solving the task, we approximate $\mathcal{H}[\mathbf{a}]$ the following way: We first quantized each selected action into $100 \times |\mathcal{A}|$ bins, where $\mathcal{A}$ is the action space. We then calculated a categorical distribution over actions based on the frequencies of the quantized actions, the entropy of which we used as our approximation for $\mathcal{H}[\mathbf{a}]$. The categorical distribution was calculated based on actions selected over 50 episodes.

### D.1 Return per bit for stochastic policies

We evaluated the information efficiency of the stochastic variants of the policies learned by LZ-SAC, SPAC, SAC and MIRACLE. We approximated the entropy of the distribution of actions in the same way described above, sampling actions over 50 episodes. To compute the conditional entropy of actions given the state $\mathcal{H}[\mathbf{a}|\mathbf{s}]$, we sampled 1000 actions from the policy at every state $\mathbf{s}_t$. We then calculated a categorical distribution (described in previous section) based on this sample, the entropy of which we used as our approximation of the conditional entropy $\mathcal{H}[\mathbf{a}|\mathbf{s}]$. In this setting too, the SPAC algorithm tends to produce the most information efficient agents (Fig. 10).

## E Open-loop control

Increasing the number of closed-loop actions used to prompt the transformer makes it generate more rewarding action sequences. This shows the importance of providing the sequence models with enough context, to be able to predict rewarding behaviors (Fig. 11).

## F Partial observability

The augmented reward function that induces the preference for simple action sequences depends on the actions the agent selected in the past. This makes the reward function partially observable for a purely state-conditioned policy. Our agents learn to maximize this reward function despite this partial observability. We tested whether augmenting the state to contain information about actions selected in the past produced substantial differences in the learned policies. We equipped the LZ-SAC agents with a recurrent neural network (a Gated Recurrent Unit [55]) whose inputs were sequences of actions. We trained this network along with a single readout layer to produce embeddings $\mathbf{e}_t$ of action sequences with which we defined the augmented state $\mathbf{a}_t \sim \pi_\theta(\cdot|\tilde{\mathbf{s}}_t)$ where

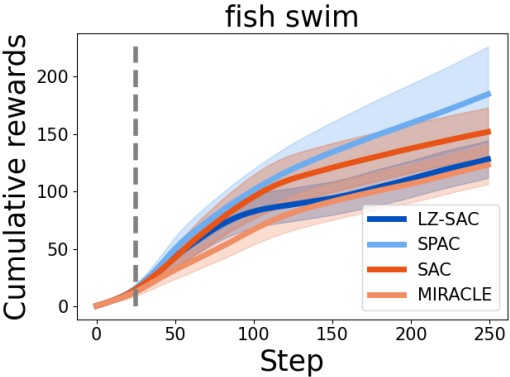

Figure 11: Cumulative rewards attained by sequence priors in the `fish swim` environment, with a prompt length of 25.

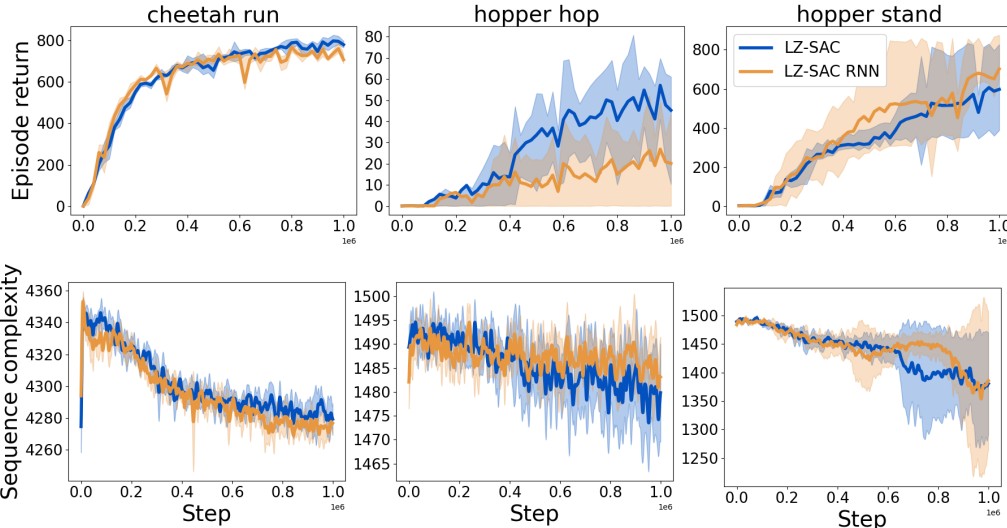

Figure 12: Returns and compressibility of action sequences when the reward function is partially observable and fully observable.

$\tilde{s}_t = Concatenate(s_t, e_t)$. In three tasks we observed only minor differences in the policies learned (Fig 12).

## G  Pretraining the prior

We trained Transformer models to perform next-action prediction from action sequences produced by the converged LZ-SAC agent for all tasks. Using the pretrained transformers (with frozen weights) rather than a randomly initialized one whose weights were updated with stochastic gradient descent sped up learning significantly and allowed the SPAC agent to learn more rewarding behaviors (see Fig. 13). This showcases an interesting possible connection between our sequence compression framework and behavioral cloning.

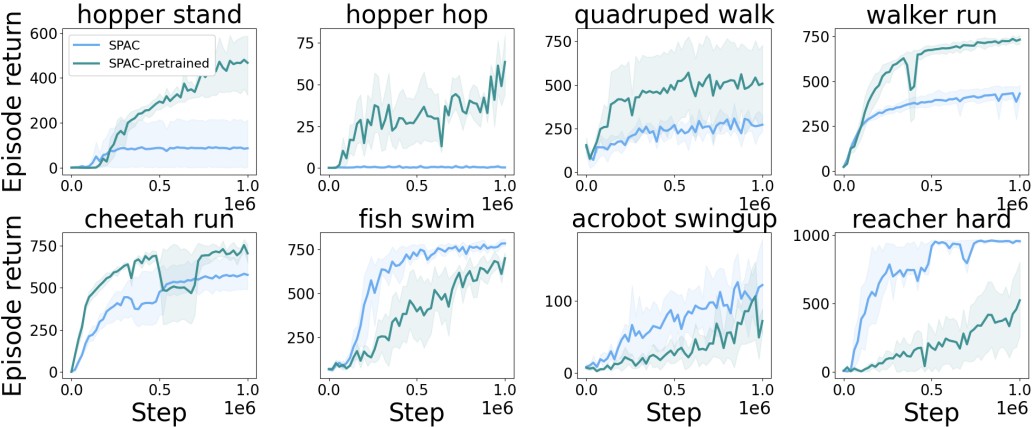

Figure 13: Learning curves for SPAC using a pretrained transformer with frozen weights and a randomly initialized transformer. Using a pretrained transformer allows SPAC to solve the more challenging tasks in the benchmark. Learning curves are averaged over three seeds. Shaded region represents 20-80 percentile.

