# OpenReview forum: "Reinforcement Learning with Simple Sequence Priors"
_NeurIPS.cc/2023/Conference — NeurIPS 2023 poster_

### Official Review · Reviewer_7uiK · 2023-06-29

**Soundness:** 3 good
**Presentation:** 3 good
**Contribution:** 3 good
**Rating:** 6
**Confidence:** 3

**Summary:**

This work proposes to use action-based complexity cost. This work argues that this action-based cost can be formulated either as compression over the experienced so far actions or using a transformer that predicts the next action given the past. This approach is then claimed to provide access to simplified / predictable action sequences. The work is evaluated against a number of dmcontrol tasks and is shown to do better than considered alternatives, both in terms of performance but also in terms of learning policies that are easier to learn. Overall, the work proposes an interesting set of evaluation and presents an interesting idea. However, I found the work fairly difficult to follow at times, whereby paper clarity can be improved. I am also unsure that the proposed solution is necessarily going to be useful, given the presence of theoretically stable alternatives, such as [4]. Finally, I think this work has potential but should be evaluated on more complex tasks - such as tasks from the field of robotics and / or tasks where exploration / initially bootstrapping policy learning is important. As a result, I cannot recommend this work for publication yet.

**Strengths:**

- A novel action sequence - driven reward formulation

- Interesting and informative experimentation

- Good literature review coverage

**Weaknesses:**

- No complex experiments that can further strengthen the claims for policy search and robustness

- Insufficient comparisons to prior works, e.g. such as [4] but also works like HER

- Some unclear prose, specifically the abstract, introduction and methods section can benefit from less grandous wording

- Additional justification of the choice of transformer (as opposed to something simpler) and also some argumentation as to why this cannot be pre-trained ahead of time and then fixed during training

**Questions:**

Nothing explicit.

**Limitations:**

The limitations section is a bit lacking, it is unclear how this would work in the context of complex sequential / long-horizon tasks. Also, it is unclear how would this approach work on policies learnt from pixels.

---

> ### Author Rebuttal · Authors · 2023-08-09
>
> We thank the reviewer for the constructive and thoughtful feedback. We are glad that the reviewer found our work interesting, and we are encouraged by the fact that the reviewer thinks it has potential. The reviewer points out four main limitations of the submitted paper. We have addressed all four weaknesses pointed out by the reviewer, and it has made our paper a lot stronger. We address the weaknesses in detail below:
>
> >No complex experiments that can further strengthen the claims for policy search and robustness. [...] Also, it is unclear how would this approach work on policies learnt from pixels.
>
> The reviewer raised the point that we do not address how our algorithm might work in pixel-based environments. We have since added experiments showing that LZ-SAC can be adapted to pixel-based tasks too. We trained LZ-SAC and SAC to perform control from pixels in the DeepMind Control Suite 100k benchmark (DMC 100k), where the agents interact with the task through 100k environment steps. To adapt LZ-SAC to the visual control domain, we equipped it with a convolutional neural network encoder and performed a random shift augmentation on batches of images when we trained the actor and critic. We adapted SAC the same way to make the methods comparable. We see that, with our simple modification and compression bonus, LZ-SAC can beat recent approaches like CURL and SAC+AE, as well as our modified SAC implementation, on average across the six visual control tasks from DMC100k (see Fig 2.B and Table 1. in the one-page pdf).
>
> >Finally, I think this work has potential but should be evaluated on more complex tasks - such as tasks from the field of robotics and / or tasks where exploration / initially bootstrapping policy learning is important.
>
> To increase the breadth of tasks that we use for evaluating LZ-SAC, we included three robotic manipulation tasks from the metaworld benchmark. Across the three tasks we see that LZ-SAC learns to solve the manipulation problems faster and more consistently than SAC (Fig 2.A in the one-page pdf). Moreover, the action sequences LZ-SAC uses to solve these tasks do not exhibit periodic properties, but are still smooth and predictable (Fig 2.C in the one-page pdf). This showcases the possibility of using LZ-SAC not only in tasks with strong periodic patterns, but also in more generic continuous control settings.
>
> >Insufficient comparisons to prior works, e.g. such as [4] but also works like HER
>
> We implemented Robust Predictable Control (RPC) (e.g. ref [4]) and evaluated it on all the DeepMind Control Suite tasks from the original submission. We found that RPC could not solve the tasks with the same information budget as our other models, and increased the budget to make RPC more performant. Still with this fine-tuning RPC was outperformed by both LZ-SAC or SPAC. Both of our methods are therefore strong alternatives to existing baselines. We note that prior works like HER are orthogonal to the methods proposed in this paper, and can therefore be combined with our approach.
>
> >Some unclear prose, specifically the abstract, introduction and methods section can benefit from less grandous wording
>
> We have cleaned up the prose at various places throughout the manuscript, including the abstract and introduction. See our changes below. Strikes indicate that the text is removed.
>
> **"~~Everything else being equal, simpler models should be preferred over more complex ones.~~ In reinforcement learning (RL), simplicity is typically quantified on an action-by-action basis [...]"**
>
> **"Simplicity is an ~~powerful~~ important inductive bias. In science, we strive to build parsimonious simple theories and algorithms that involve repetitions of the same basic steps."**
>
> **"~~Though we want our RL agent to maximize rewards, we encourage it to do so with policies that produce simple action sequences.~~ We train our agents to learn policies that maximize reward as well as the compressibility of the action sequences they produce."**
>
> >Additional justification of the choice of transformer (as opposed to something simpler) and also some argumentation as to why this cannot be pre-trained ahead of time and then fixed during training
>
> We would like to thank the reviewer for suggesting pretraining the transformer ahead of time. We trained transformer models to perform next-action prediction from action sequences produced by the converged LZ-SAC agent for all tasks. Using the pretrained transformers (with frozen weights) rather than a randomly initialized one sped up learning significantly and allowed the SPAC agent to learn more rewarding behaviors (see Fig 4 in the one-page pdf). This showcases an interesting possible connection between our sequence compression framework and behavioral cloning.
>
> Lastly, we added a justification of our use of the transformer rather than a simpler model. While it is perfectly possible to combine our method with a simpler sequence prior, transformers are the SOTA models for learning complex sequence data, and are better suited for modeling long-range dependencies. We have added the following justification to our paper:
>
> **"We use transformers rather than simpler architectures since they are better suited for  learning complex sequence data with long-range dependencies, as we expect to see in an RL setting."**

---

> > ### Comment · Reviewer_7uiK · 2023-08-10
> > **Thanks for the detailed rebuttal, well done**
> >
> > Dear authors,
> >
> > I would like to start by thanking you for the detailed and thorough rebuttal to my comments, I am glad you found them useful! I will be happy to increase my score provided I see the referenced by you 1 page long pdf file and the reported updates here match the results there.
> >
> > This document is nowhere to be found, perhaps it is set to different visibility or it is not submitted yet.
> >
> > Thank you once more, great job!

---

> > > ### Comment · Reviewer_6AwJ · 2023-08-10
> > > **Rebuttal 1-page PDF**
> > >
> > > Hi Reviewer 7uiK,
> > >
> > > Are you referring to the PDF linked at the end of the global author rebuttal at the top? I've copied the link here for more visibility: [https://openreview.net/attachment?id=v1lUnN8xAL&name=pdf](https://openreview.net/attachment?id=v1lUnN8xAL&name=pdf)

---

> > > ### Author Response · Authors · 2023-08-10
> > > **Thank you! The pdf should be visible now**
> > >
> > > Dear Reviewer 7uiK,
> > >
> > > We are thankful that you liked our rebuttal and that you are willing to increase your score. The one-pager was uploaded to the global response and there is a link to download it at the very bottom of the global response. The one-pager is visible in our console, and we hope that is visible to all reviewers as well.
> > >
> > > Thank you to Reviewer 6Awj for providing a link to the one-pager. We hope that the platform is displaying our response properly but please let us know in case anything is unclear.
> > >
> > > Again, we appreciate that the reviewer thinks we did a good job improving the paper: This would not have been possible without the instructive feedback.

---

> > > > ### Comment · Reviewer_7uiK · 2023-08-10
> > > > **Thanks both, score updated**
> > > >
> > > > Dear authors and reviewer 6Awj, thanks for the link, I had completely overlooked it.
> > > >
> > > > After reviewing the reported results along with the provided rebuttal I am happy to update my score.
> > > >
> > > > Thanks once more for the careful and thorough replies!

---

> > > > > ### Author Response · Authors · 2023-08-14
> > > > > **Thank you again**
> > > > >
> > > > > We would like to thank you again for engaging with our paper during the rebuttal, and for raising your score. It was encouraging to see that our experiments and clarifications to the paper could make a difference.

---

### Official Review · Reviewer_GvQF · 2023-06-30

**Soundness:** 1 poor
**Presentation:** 2 fair
**Contribution:** 2 fair
**Rating:** 5
**Confidence:** 4

**Summary:**

This paper proposes the idea of using simplicity as a prior when learning control policies using RL. Simplicity (or complexity) here is defined as the cost of predicting action $a_t$ given past actions $(a_\{t - \tau:t-1})$ and current state ($s_t$). Two different approaches are proposed to learn this complexity value. First, using a learned neural network that learns to predict current action given past actions trajectory $(\phi(a_t | a_{t-\tau:t-1}$. This is used as additional reward in SAC.

Second approach, aims to use lossless data compression algorithms (lz4) and the motivation is to compute complexity as the additional number of bits required to encode $a_t$ given that we have already encoded  $a_{t-\tau:t-1}$.

**Strengths:**

Overall, the paper is well written and accessible. The idea of simplicity as a prior is also interesting.

**Weaknesses:**

I think the paper is very vague in terms of using concepts such as simplicity. In the Related works section, the paper simply connects simplicity with maximum entropy RL and suggests that a uniform policy is simple (”stay close to simple uniform prior policy”). I don’t see this connection, infact I a uniform random policy leads to random walks and infact maybe highly non-simple (see e.g. brownian motion and discontinuous behavior).

On the approach side the first approach to predict actions from past is very strange since it results in trying to model non-stationary behavior and thus makes little sense. This is sort of elluded in the paper as well, where the paper suggests that learning a policy with this reward can be challenging.  From the results we can further see that this approach really doesn’t work.

The idea of using LZ4 would try to reduce new actions thus sort of minimizing surprise. However, interestingly, while this approach performs well it doesn’t perform unifromly well across all tasks e.g. on 5/8 tasks (Figure 5) it performs quite poorly.

I also find some baseline SAC  numbers to be strange e.g. hopper-hop, quadraped-walk, walker-run have all been shown to be solved using SAC. However, the numbers reported in this paper are very low. I don’t think these baseline numbers are fully accurate or in case there were some other differences maybe the authors can comment.

Related Works: There are many other approaches in the literature which focus on simplicity by compressing action sequences into latent variables, or using surprise minimization  as additional part of the objective. None of these approaches are referred to or compared in the text. I would recommend the authors to cite them, and compare against them as well.

Achiam et al. Surprise-based intrinsic motivation for deep reinforcement learning

Berseth et al. SMIRL: Surprise Minimizing Reinforcement Learning in Unstable Environments

**Questions:**

Do the authors know why the baseline numbers (SAC) for these simple environments are very low?

**Limitations:**

limitations are discussed in the main paper.

---

> ### Author Rebuttal · Authors · 2023-08-08
>
> We thank the reviewer for the detailed and thorough feedback, and we are glad that the reviewer found our idea of using sequence simplicity as a policy prior in RL interesting. In the following we address the issues raised by the reviewer: We have clarified what we mean by simplicity, as well as the strengths and weaknesses of SPAC. We have also fine-tuned SAC to be more performant on Hopper Hop and Quadruped Walk. Lastly, we conducted an experiment where we pretrain the SPAC transformer on converged policies, which improves performance. We thank the reviewer for raising these issues. We think our paper’s claims are better supported as a result of the reviewer’s feedback.
>
> >On the approach side the first approach to predict actions from past is very strange since it results in trying to model non-stationary behavior
>
> The reviewer is correct to point out the challenges of training the transformer towards non-stationary targets. However, the training is intended to encourage the agent to solve tasks with predictable and simple action sequences. We implemented another baseline using an information bottleneck, Robust Predictable Control (RPC), that uses a learnable prior over states to compress state-representations. In our experiments SPAC outperforms RPC on average over the 8 DeepMind control tasks (see Fig 1.A and Fig 1.B in the one-page pdf). This shows that, for a model with a strong information bottleneck, SPAC is quite performant.
>
> To address the issue of non-stationarity further, we pretrained the transformer model to predict the action sequences of converged policies, and trained the SPAC agent using this pretrained transformer, instead of a randomly initialized one, on all tasks from the original submission. This agent learns to solve the tasks more effectively, attaining scores comparable to SAC in challenging domains like walker and quadruped (see Fig. 4 in the one-page pdf).
>
> >I also find some baseline SAC numbers to be strange e.g. hopper-hop, quadraped-walk, walker-run have all been shown to be solved using SAC.
>
> We thank the reviewer for pointing this out. We repeated the experiments for Hopper Hop and Quadruped Walk with larger network sizes and lower information costs, and now obtain scores comparable to those reported in other published papers [1, 2]. We have updated the scores we report in the paper to reflect this, see Fig 1.A in the one-page pdf for the updated scores. The SAC agent does indeed solve the walker run task, and the score was always consistent with those reported elsewhere [1, 2]. Our LZ-SAC algorithm still outcompetes SAC after fine-tuning.
>
> >while this approach performs well it doesn’t perform unifromly well across all tasks e.g. on 5/8 tasks (Figure 5) it performs quite poorly.
>
> The experiment presented in Figure 5 is not the main result of our paper, rather it shows that the models we evaluated solve the tasks using various amounts of bits of information. We find that SPAC achieves the highest return per bit of information it uses to solve the tasks. While LZ-SAC is better at maximizing reward, it uses more information to do so. SPAC therefore, we argue, is a strong algorithm for performing information-constrained control.
>
> >There are many other approaches in the literature which focus on simplicity by compressing action sequences into latent variables, or using surprise minimization as additional part of the objective.
>
> We appreciate the reviewer’s pointers to related work. We updated the related work section to discuss similarities to our approach (see below). While Berseth et al. learn a policy that maximizes the predictability of the next state given the current state, we maximize the predictability/compressability of the next action, given previous actions. Moreover, we evaluated our algorithms in the DeepMind Control Suite, which does not satisfy the criterion of being an unstable environment as described in Berseth et al. In these environments, the agent could easily maximize predictability by not moving, remaining in the same state throughout the episode.
>
> **"Related to compression is predictability: Berseth et al. [21 ] learn a density model over states, and then learn a policy that seeks out states that are predictable, leading to self-sustaining behaviors in unstable environments. On the opposite end there are methods that seek out unpredictable states [22, 23], or states that the agent cannot compress, to improve exploration."**
>
> >the paper simply connects simplicity with maximum entropy RL and suggests that a uniform policy is simple (”stay close to simple uniform prior policy”). I don’t see this connection, infact I a uniform random policy leads to random walks and infact maybe highly non-simple
>
> The reviewer raises an interesting point that maximum entropy RL can produce random behavior that is not predictable. However, as is common in the RL literature  [3, 4], we understand the simplicity of an input-output relation as the mutual information between the states and the actions. Since a uniform random policy does not use information about the state to select actions, it is considered simple. Through the lens of mutual information minimization, there is therefore a clear link between our surprise minimization and maximum entropy RL. We have rewritten parts of the exposition to make this connection clearer (see below).
>
> **"Though uniform priors can lead to discontinuous and unpredictable behaviors, maximum entropy methods are considered simple in that they try to minimize the use of information about the state to select actions [4, 15, 16]."**
>
> [1] Eberhardt et al. 2023 ICLR. Pink noise is all you need: Colored noise exploration in Deep Reinforcement Learning
>
> [2] Hansen et al. 2022 ICML. Temporal Difference Learning for Model Predictive Control
>
> [3] Eysenbach et al. 2021 NeurIPS. Robust Predictable Control
>
> [4] Bassily et al. 2018 Algorithmic Learning Theory. Learners that Use Little Information

---

> > ### Comment · Reviewer_GvQF · 2023-08-12
> >
> > Thank you for the detailed response.
> >
> > > .. report in the paper to reflect this, see Fig 1.A
> >
> > Looking at the results it seems that only on Hopper-Hop and Quadraped-Walk is the method much better? On other environments the performance of SAC seems quite comparable.  Further, LZ-Spac seems to suffer from high variance.
> >
> > > Through the lens of mutual information minimization
> >
> > Thank you, I appreciate the clarification.
> >
> > Overall, I have updated my score to 5. I still think the overall approach is not very principled. It's also not very clear to me why this approach will perform better in environments that do not have any periodicity (most dm_control tasks considered here have periodicity).
> >
> > Finally, works that explicitly take into account periodicity of an agent could also be useful to cite and compare against.
> >
> > Sharma et al. Phase-Parametric Policies for Reinforcement Learning in Cyclic Environments

---

> > > ### Author Response · Authors · 2023-08-14
> > > **Thank you for the detailed feedback**
> > >
> > > Dear Reviewer GvQF,
> > >
> > > Thank you for actively taking part in the rebuttal process and for continuing to provide us with feedback. We are glad that you found our additions to the paper clarifying, and that you have increased your score.
> > >
> > > >It's also not very clear to me why this approach will perform better in environments that do not have any periodicity (most dm_control tasks considered here have periodicity).
> > >
> > > Several reviewers  had also raised this point. To further address this question, we evaluated LZ-SAC against SAC in three robotic manipulation tasks from metaworld, where we expect periodicities to be a lot less prevalent. Our results show that LZ-SAC learned to solve the tasks faster (see Fig 2.A in the one-page pdf). While these tasks do not contain periodicities, LZ-SAC prefered to solve them with smooth and predictable motions (see Fig 2.C).
> > >
> > > >Sharma et al. Phase-Parametric Policies for Reinforcement Learning in Cyclic Environments
> > >
> > > Thank you for the suggestion, we now reference this work when introducing the DeepMind Control Suite:
> > >
> > > **“We evaluated the agents described in Section 3 on eight continuous control tasks from the DeepMind Control Suite [ 34 ]. Many of these tasks promote behaviors with periodic elements, such as running and walking. While specialized architectures exist for such tasks [ 35 ], we expect compressibility to be a useful inductive bias for learning these behaviors.”**
> > >
> > > Lastly, we would like to state that we believe the framework we use in our paper is indeed appropriately principled. LZ-SAC and SPAC maximize the sum of discounted rewards minus a tractable bound on the mutual information between sequences of states, and sequences of actions. Such information-constrained formulations of the learning objective show up in the control [1], cognitive science [2] and neuroscience [3] literature.
> > >
> > > We thank the reviewer again for engaging with our paper so thoroughly during the review process, and for the feedback and fruitful suggestions made. We think that our paper has improved a lot by addressing the points raised by the reviewer.
> > >
> > > [1] Eysenbach et al. 2021 NeurIPS, Robust Predictable Control.
> > >
> > > [2] Bhui et al. Resource-Rational Decision Making 2021. Current Opinion in Behavioral Sciences.
> > >
> > > [3] Zador 2019 Nature Communications. A critique of pure learning and what artificial neural networks can learn from animal brains.

---

### Official Review · Reviewer_6AwJ · 2023-07-03

**Soundness:** 3 good
**Presentation:** 3 good
**Contribution:** 3 good
**Rating:** 7
**Confidence:** 4

**Summary:**

This paper proposes two approaches to regularizing RL policies based on sequential action priors, rather than on single action priors. One approach is SPAC, which trains an autoregressive open-loop policy prior and uses it as a reference regularizer. Another approach uses the LZ4 compression algorithm to estimate how compressible an action sequence is, and uses that as the regularizer. It tests both approaches along with two baselines, SAC and MIRACLE across a few DM Control tasks, and show that the learned policies end up being simpler (more compressible), as well as more performant.

**Strengths:**

The main strength of the paper lies in the novelty of using LZ4 for a prior and its results. A priori, it is not clear at all whether relying on a dictionary-based compression method that is essentially doing substring matching would translate well to an RL domain, and the results are novel and interesting to see. Including SPAC as well helps to round out the results, showing that a more generic approach to sequence priors is possible. The method and results are mostly quite clear.

**Weaknesses:**

The ablations for noise and open-loop policies are maybe not the most pertinent ablations to have in the main paper.

Perhaps there should be ablation for the size and architecture of the transformer prior for SPAC, as a potential way of controlling the complexity/simplicity of the policy prior. It would’ve shed more light on the tradeoff between simplicity vs. performance.

Similarly, an ablation for how the actions are compressed with LZ4 (there really should be a short explanation in the main paper about this) such as changing the resolution for discretization would’ve also been very enlightening, to again show the tradeoff between simplicity and performance.

Finally it would’ve been very useful to include a different kind of domain, perhaps gridworld-based domains (MinAtar/Atari) to see whether we can see similar trends.

---- After Author Rebuttal ----
I appreciate the additional ablations for the transformer architecture and action discretization. Therefore I have increased my score.

**Questions:**

Did all four agents use the same SAC that was augmented with multiple past actions? Or was the augmented SAC only used for LZ-SAC and SPAC? If this is not the case, then the same augmented SAC should be used for all four agents to be consistent.

Robustness to noise ablation: Were the agents retrained with the gaussian noise, or was the noise added to the already-trained agents? The plot shows all agents performance decaying relatively similarly, so it doesn’t seem like LZ-SAC or SPAC were particularly more robust to noise. Perhaps the details of this ablation is better put in the appendix so a different ablation can be in the main paper, such as over the transformer architecture or action discretization for LZ4.


**Limitations:**

The paper already discusses many of the limitations/weaknesses brought up earlier.

---

> ### Author Rebuttal · Authors · 2023-08-08
>
> We would like to thank the reviewer for the helpful feedback and the encouraging review. We particularly appreciate that the reviewer found our method and results novel and interesting - both our use of dictionary-based compression and generic compression with sequence models. We have focused on running the ablation experiments the reviewer suggested, as well as adding more evaluations of our method.
>
> >Perhaps there should be ablation for the size and architecture of the transformer prior for SPAC, as a potential way of controlling the complexity/simplicity of the policy prior.
>
> We ran an ablation experiment where we varied the number of heads, number of layers, and embedding dimensions of the transformer SPAC used to predict its own actions. We tested the ablated models on three DM Control domains (Cheetah, Walker and Acrobot). We find that smaller transformer architectures work well for policy regularization (see Fig 3.B in the one-page pdf), outperforming the bigger architectures. Otherwise we do not see a consistent benefit of larger architectures. Instead, it is possible that the extra number of parameters of the bigger architectures makes it more challenging to learn both a sequence prior and a policy concurrently. We have added the ablation experiments to the Appendix.
>
> >Similarly, an ablation for how the actions are compressed with LZ4 (there really should be a short explanation in the main paper about this) such as changing the resolution for discretization would’ve also been very enlightening
>
> We thank the reviewer for this interesting suggestion. We varied the resolution with which we discretized the action sequences used as input to the compression algorithm. Across three DeepMind Control tasks, we observe that both too low and too high resolutions remove the performance gain of LZ–SAC: When the resolution is 0, the compression bonus no longer conveys a signal about the simplicity of the policy. Conversely, if the resolution is too high, every action sequence is equally incompressible due to the continuous nature of the action space. We find that rounding to two decimal places gave the best performance on average across the tasks (Fig 3.A in the one-page pdf). We also added an extra paragraph explaining how LZ4 works in the methods section (see below), and refer to pseudo-code for LZ4 which we have in the appendix A2.
>
> **"The LZ4 algorithm compresses sequences by replacing repeating sub-sequences in the data with references to an earlier occurring copy of the sub-sequence. These copies are maintained in a sliding window. Repeating sub-sequences are encoded as *length-distance* pairs $(l, d)$, specifying that a set of $l$ symbols have a match $d$ symbols back in the uncompressed sequence. This allows the sequence to be encoded with fewer bits, should it contain such repeating sub-sequences. See Appendix A2 for pseudocode."**
>
> >Finally it would’ve been very useful to include a different kind of domain, perhaps gridworld-based domains (MinAtar/Atari) to see whether we can see similar trends.
>
> Following the suggestion of the reviewer, we have included more tasks and baselines in our evaluation of the LZ-SAC algorithm. We now include evaluations in pixel-based versions of the DeepMind Control Suite (using the sample efficient 100k benchmark), as well as from the Metaworld benchmark, containing robotic manipulation tasks. We have added results from both benchmarks to our paper.
>
> In the pixel-domain, we implement LZ-SAC and SAC like in the state-based tasks, but with a convolutional neural network encoder and applying a random shift augmentation to the images before training the actor and critic. Using this simple modification with our compression objective, we see an improvement not only over SAC, but also over recent approaches specialized for solving visual control tasks, like CURL and SAC+AE: Across six visual control tasks from the DeepMind Control Suite, LZ-SAC attains the highest average score (see Fig 2.B and Table 1 in the one-page pdf for model comparisons), showcasing the viability of using LZ-SAC in the pixel domain.
>
> Lastly we compared LZ-SAC and SAC in three metaworld tasks. Across these robotic manipulation tasks, we see that LZ-SAC learns a successful policy faster and more reliably than SAC (Fig 2.A in the one-page pdf). Furthermore, the LZ-SAC policy produced smoother, more predictable action sequences than the policy SAC learned (Fig 2.C in the one-page pdf). This suggests that using dictionary-based compressibility as a prior could be useful in more generic continuous control settings, not just in periodic motor control tasks.
>
> >Did all four agents use the same SAC that was augmented with multiple past actions?
>
> We did not find it necessary to augment the state with the past actions of the agent in the DMC experiments (see confirmatory experiments in appendix F). As such, all agents had the same input. We have added a sentence to clarify this in the paper (see below).
>
> **"We also did not find it necessary to augment the state representations of our methods with the past actions to solve the DeepMind Control Suite tasks (see Appendix F)."**
>
>
> >Were the agents retrained with the gaussian noise, or was the noise added to the already-trained agents [...] The plot shows all agents performance decaying relatively similarly, so it doesn’t seem like LZ-SAC or SPAC were particularly more robust to noise. Perhaps the details of this ablation is better put in the appendix so a different ablation can be in the main paper
>
>
> In the noise ablation, the noise was added to the already trained agents. We found the reviewer’s proposed discretization resolution ablation very interesting and we have decided to replace the noise ablation with the discretization ablation in the paper.

---

> > ### Comment · Reviewer_6AwJ · 2023-08-14
> >
> > Thank you for the additional ablations and answer to my questions! I have increased my score.

---

> > > ### Author Response · Authors · 2023-08-14
> > > **Thank you**
> > >
> > > Dear Reviewer 6Awj,
> > >
> > > Thank you again for your instructive feedback and engaging in the rebuttal. We are happy to see the score increase as a result of the new ablations and clarifications - thank you!

---

### Official Review · Reviewer_teDV · 2023-07-07

**Soundness:** 3 good
**Presentation:** 3 good
**Contribution:** 3 good
**Rating:** 6
**Confidence:** 4

**Summary:**

In this paper, the authors propose a reinforcement learning method that produces simplified action sequences. Simplicity is defined as the predictability of the next action taken by the RL agent and is measured using the number of bits required to encode the action sequence. The authors propose two methods to introduce an information bottleneck for biasing/regularizing the RL agent toward simplicity. The first method SPAC, uses a learned sequence model that predicts the next action and the RL agent is incentivized to produce predictable actions. A second method LZ-SAC directly uses a compression algorithm to compress the action sequence and penalizes the number of bits required for storage. The authors show that this regularization can have several benefits and can lead to better overall performance as compared to other action regularization techniques.


**Strengths:**

The paper is well-written and presented. The motivation provided for the simplicity of action sequences is reasonable and the overall idea is interesting for the community. Several different types of experiments have been carried out, comparing performance, performance per bit of information and the ability to use open-loop control. If validated with more experiments, the proposed LZ-SAC method could be a good way to regularize RL policies. The future ability of RL agents to discard information about the state of the world and simply use prior actions instead is promising.

**Weaknesses:**

The major weakness of the work would be the lack of further baselines. Since the approach is very general, it needs to be validated by comparison with more SOTA off-policy RL baselines to ground the results. It would also be interesting to compare against other information bottleneck methods such as [4].
It would also be beneficial if the authors try to categorize which kinds of environments benefit from compression. It could be that environments without periodicity are worse off with compression. The authors claim that the method never performs worse than SAC but this claim needs to be validated by testing on many more non-periodic tasks.


**Questions:**

- What is the methodology for choosing the compression method LZ4 over others?
- Why doesn't any agent do well in the hopper-hop task?
In section 7, how much does the performance drop in open-loop control as compared to closed-loop control? It is not clear from the axis in Figure 7. Are the differences in the cumulative rewards between the 4 methods statistically significant?
- Figure 10: Why does the LZ-SAC method work the best in this case with stochastic policies, while SPAC performs much worse? Why does stochasticity have this effect?

**Limitations:**

Yes they are addressed.

---

> ### Author Rebuttal · Authors · 2023-08-08
>
> We thank the reviewer for their thoughtful comments and feedback. We are happy that the reviewer found our idea of using action sequence compression interesting for the RL community. Moreover, it is encouraging that the reviewer acknowledges the possible impact our method could have for regularizing RL policies, if it is tested with more experiments. We conducted a series of new experiments in order to address three weaknesses that were outlined by the reviewer: First, we tested the models on tasks that are not periodic in nature (metaworld). Secondly, we tested LZ-SAC on pixel-based versions of tasks from the Deepmind Control Suite. Lastly, we implemented Robust Predictable Control (RPC) and compared it to our methods in all Deepmind Control tasks from the original submission. In total, we introduce tasks from two new benchmarks and three new baseline models  (RPC, CURL, and SAC+AE). We believe this has made our contribution more valuable, and we thank the reviewer for the suggestions. We discuss the experiments in more detail below:
>
> >The major weakness of the work would be the lack of further baselines. [...] It would also be interesting to compare against other information bottleneck methods such as [4].
>
> Since several reviewers suggested it, we implemented Robust Predictable Control (RPC) (from [4]) and tested it on all DeepMind Control suite tasks from the original submission. We had to increase the information budget of RPC to make it performant in our task, and even then it is outperformed both by LZ-SAC and SPAC on average (see Fig 1.A and 1.B in the one-page pdf). Our method is thus a viable alternative to existing baselines.
>
> >It would also be beneficial if the authors try to categorize which kinds of environments benefit from compression. It could be that environments without periodicity are worse off with compression.
>
> To make a stronger case for our method, we tested LZ-SAC on more tasks. First, we trained our algorithm on pixel-based versions of tasks from the Deepmind Control Suite. We tested LZ-SAC in the sample efficiency domain, where the agent experiences 100k environment steps. We modified LZ-SAC with a convolutional neural network encoder and performed a random shift augmentation to images before training the actor and critic. With this simple modification, LZ-SAC outperforms not only SAC with the same modification, but also SOTA off-policy models like CURL and SAC+AE on average (see Figure 2.B and Table 1 in the one-page pdf). Simple action sequence priors can therefore be useful for solving high-dimensional visual control tasks.
>
> Second, we trained LZ-SAC and SAC on three robotic manipulation tasks from the metaworld benchmark, where we expected periodic sequences to play far less of a role. LZ-SAC not only learns to solve the tasks faster than SAC, but more consistently too (Fig 2.A in the one-page pdf). Moreover, LZ-SAC does not solve the tasks with periodic action sequences, but rather with smooth and predictable ones (see Fig 2.C in the one-page pdf).
>
> >The authors claim that the method never performs worse than SAC but this claim needs to be validated by testing on many more non-periodic tasks.
>
> We do not wish to claim that LZ-SAC never performs worse than SAC (just that it was equal or better in our experiments), and we have added the following sentence to the results section:
>
> **"Though we do not expect LZ-SAC to always outperform SAC in more generic control settings, we see an improvement in many tasks with periodic elements, like walking and running."**
>
> What is the methodology for choosing the compression method LZ4 over others?
>
> >We used LZ4 due to its fast compression speed. This information is now added in the methods section in the paper (see below). We justified this further by testing LZ-SAC with two other lossless compression algorithms (bzip and zlib) on the Cheetah - Run task, and found no significant difference in performance (see Fig 3.C in the one-page pdf).
>
> **"We chose the LZ4 algorithm due to its state-of-the-art compression speed (see Appendix G for experiments with other compression algorithms)"**.
>
> >Why doesn't any agent do well in the hopper-hop task?
>
> We thank the reviewer for pointing this out. We repeated the experiments for Hopper-Hop with larger network sizes and lower information costs, and see that the SAC scores are consistent with other papers [1, 2]. To make the comparison fair we make the same changes to LZ-SAC and see that it remains superior to SAC. We have added the new scores to the paper (Fig 1.A in the one-page pdf).
>
> >In section 7, how much does the performance drop in open-loop control as compared to closed-loop control? [...] Are the differences in the cumulative rewards between the 4 methods statistically significant?
>
> The performance drops vary across tasks. In tasks like Cheetah-Run and Walker-Run, SPAC and LZ-SAC achieve roughly 10% of the closed-loop performance, only observing the first 5% of the states in the episode.  Conducting t-tests, we find that SPAC scores (averaged over tasks) for open-loop control are significantly higher than both LZ-SAC (p=0.006), SAC (p=0.004) and MIRACLE scores (p=0.001). We have added these statistics to the paper.
>
> >Figure 10: Why does the LZ-SAC method work the best in this case with stochastic policies, while SPAC performs much worse? Why does stochasticity have this effect?
>
> This decrease in the return per bit ratio does not reflect a decrease in returns for SPAC, but rather that our approximation of the mutual information between states and actions is lower for the stochastic LZ-SAC policy. Since we do not have analytic expressions for the entropy of the action marginal or policy, we approximate this through sampling (more information in Appendix D).
>
> [1] Eberhardt et al. 2023 ICLR. Pink noise is all you need: Colored noise exploration in Deep Reinforcement Learning
>
> [2] Hansen et al. 2022 ICML. Temporal Difference Learning for Model Predictive Control

---

> > ### Comment · Reviewer_teDV · 2023-08-11
> >
> > I thank the reviewers for the detailed answers and for conducting further experiments. I believe the additions of new results and clarifications to the main paper make the work worthy of publication, therefore, I am happy to increase my score.

---

> > > ### Author Response · Authors · 2023-08-14
> > > **Thank you**
> > >
> > > Dear Reviewer teDV,
> > >
> > > We are grateful for the time you spent reviewing and engaging with our rebuttal. We are happy that the inclusion of new results and additional clarifications have made you think our paper is worthy of publication and that you have increased your score. We would like to thank you again for actively taking part in the reviewing process and providing such constructive feedback.

---

### Author Rebuttal · Authors · 2023-08-09

We would like to thank all of the reviewers for the time and effort put into providing thoughtful feedback on our paper.

* Reviewer teDV found the paper “well-written and presented” and our idea “interesting for the community.”
* Reviewer 6AwJ praised our paper for its “novelty of using LZ4 for a prior and its results.”
* Reviewer GvQF said that the “paper is well written and accessible” and that the “idea of simplicity as a prior is also interesting.”
* Reviewer 7uiK mentioned that our “work proposes an interesting set of evaluations and presents an interesting idea.”

However, at the same time, reviewers also made important suggestions, especially regarding additional baselines and benchmarks. We believe that we were able to incorporate all of these suggestions and believe that doing so has improved our paper significantly. To summarize, we have made the following additions:

* We evaluated our methods on nine additional benchmarks (requested by reviewers teDV, 6AwJ, 7uiK):
- Six pixel-based tasks from the DeepMind Control Suite, highlighting that our approach scales and remains competitive in high-dimensional, visual environments.
- Three robotic manipulation tasks from the Meta-World benchmark, demonstrating applicability to domains that rely less on periodicity.
* We implemented three additional baselines – Robust Predictable Control (RPC), Contrastive Unsupervised Reinforcement Learning (CURL), and Soft Actor Critic + Auto-Encoder (SAC+AE) – and found that LZ-SAC and SPAC remain superior to all of these baselines (reviewers teDV, 7uiK).
* We improved the performance of the SAC baseline by using larger network sizes and lower information costs (reviewers teDV, GvQF). Even with these changes, LZ-SAC remains superior to SAC.
* We ran additional ablations for both SPAC and LZ-SAC (reviewer 6AwJ, GvQF):
- an ablation experiment where we varied the number of heads, number of layers, and embedding dimensions of the transformer SPAC used to predict its actions.
- an ablation experiment where we varied the resolution with which we discretized the action sequences used as input to the compression algorithm.
- we pretrained the transformer model to predict the action sequences of converged policies and trained the SPAC agent using this pretrained transformer which improved performance even further.
* Incorporated references requested by the reviewers (reviewer GvQF).

Each of these changes is outlined in detail in our responses to the individual reviews below. We again want to thank the reviewers for their time and for actively taking part in the review process.

---

> ### Author Response · Authors · 2023-08-21
> **Discussion period summary**
>
> Dear all,
>
> We would like to express our gratitude to all reviewers for participating in the review process, and engaging with our rebuttal. The reviewers made important suggestions regarding further evaluations and baselines. We addressed these by comparing LZ-SAC to more baselines, like Robust Predictable Control, and by evaluating it on pixel-based DeepMind Control Suite tasks and robotic manipulation tasks from the Metaworld benchmark. We also conducted ablation experiments suggested by the reviewers. Now all reviewers recommend acceptance, with an average rating of 6. We think implementing the changes and extra evaluations suggested by the reviewers made our paper considerably stronger, and we are happy that this is reflected in the increased scores. Thank you all again!

---

### Decision · Program_Chairs · 2023-09-21

**Decision:**

Accept (poster)

**Comment:**

The paper proposes a simple modification to the reward in RL that penalizes the complexity of action sequences, measured either by the log likelihood of the next action prediction (using a transformer sequence model), or by the number of bits to compress the next action using standard compression algorithms. This is shown to perform well and produce smooth/structured action sequences in various RL domains.

Following the rebuttal, which added more experimental results, all reviewers agreed that the paper should be accepted.